# Success of community approach to HPV vaccination in school-based and non-school-based settings in Haiti

Cynthia Riviere[1], Tatiana Bell[1], Yonie Cadot[1], Christian Perodin[1], Benedict Charles[1], Claudin Bertil[1], Jazreel Cheung[2‡¤a], Shalmali Bane[2‡¤b]*, Hoi Ching Cheung[2‡], Jean William Pape[1,3], Marie Marcelle Deschamps[1]

**1** GHESKIO Center, Port-au-Prince, Haiti, **2** Analysis Group, Inc., Boston, Massachusetts, United States of America, **3** Center for Global Health, Division of Infectious Diseases, Department of Medicine, Weill Cornell Medical College, New York, New York, United States of America

☯ These authors contributed equally to this work.
¤a Current address: Center for Healthcare Quality and Analytics, Children's Hospital of Philadelphia, Philadelphia, Philadelphia, United States of America
¤b Current address: Department of Epidemiology and Population Health, Stanford University, Stanford, California, United States of America
‡ JC, SB and HCC also contributed equally to this work.
* shalmali.bane@gmail.com

**Data Availability Statement:** All relevant data are within the manuscript and its S1 Dataset and S1 File.

## Abstract

### Objectives

To assess the success of a human papillomavirus (HPV) vaccination program among adolescent girls aged 9–14 years in Haiti and to understand predictors of completion of a two-dose HPV vaccination series.

### Methods

Data collection was conducted during HPV vaccination campaigns in Port-au-Prince between August 2016 and April 2017. Descriptive statistics and logistic regression models were used to examine characteristics associated with vaccination series completion of school based and non-school based vaccination delivery modalities.

### Results

Of the 2,445 adolescent girls who participated in the awareness program, 1,994 participants (1,307 in non-school program, 687 in school program) received the first dose of the vaccine; 1,199 (92%) in the non-school program and 673 (98%) in the school program also received the second dose. Menarche (OR: 1.87; 95% CI, 1.11–3.14), if the participant was a prior patient at the GHESKIO clinics (OR: 2.17; 95% CI, 1.32–3.58), and participating in the school-based program (OR: 4.17; 95% CI, 2.14–8.12) were significantly associated with vaccination completion.

**Funding:** This project carried out at GHESKIO was supported by the MSPP, Partners in Health, and Merck Pharmaceuticals. No funding was received for the conduct of this project. The Analysis Group Inc. provided support in the form of salaries for authors [JC, SB, HC], but did not have any additional role in the study design, data collection and analysis, decision to publish, or preparation of the manuscript. The specific roles of these authors are articulated in the 'Author Contributions' section of the manuscript.

**Competing interests:** Authors JS, SB, and HC were employees of Analysis Group Inc. at the time of manuscript development. This does not alter our adherence to PLOS ONE policies on sharing data and materials.

## Conclusions

Vaccination in school- and non-school-based settings was successful, suggesting that a nationwide HPV vaccination campaign using either approach would be successful using either approach.

## Introduction

Cervical cancer is the fourth most prevalent cancer in women and the fourth most frequent cause of cancer-related deaths among women worldwide [1]. These deaths are largely concentrated in the developing world, with approximately 85% of the deaths from cervical cancer occurring in low- and middle-income countries [2]. In Haiti, cervical cancer is the second leading cause of cancer-related deaths among women, resulting in 563 deaths annually [3]. Most cases of cervical cancer are caused by infection with a high-risk strain of human papillomavirus (HPV) [4–6]. In the Caribbean region, the prevalence of the high-risk strains HPV 16/18 in women with normal cervical cytology is estimated to be 15.8% [7].

Cervical cancer can be prevented through the screening and treatment of early lesions, as well as the administration of HPV vaccine [8–10]. There are currently three HPV vaccines available: the bivalent, quadrivalent, and nanovalent vaccines [11, 12]. The World Health Organization (WHO) recommends vaccination against HPV in girls aged 9 to 14, targeting primarily those who are not yet sexually active [13]. For girls aged 9 to 14, a two-dose schedule is recommended for cost effectiveness and to facilitate higher coverage in adolescents.

The HPV vaccine has already been introduced in many developing countries, such as Rwanda, Botswana, Thailand, South Africa, Mexico, and Kenya, through a variety of delivery strategies, including health facility-based (i.e., non-school-based) and school-based programs [5, 14–20]. The Haitian Ministry of Public Health and of the Population (MSPP) considers cervical cancer a national public health priority and is willing to support the implementation of structures necessary for the introduction of the HPV vaccine in Haiti, including the launch of a nationally accessible screening program.

In 2009, a three-dose HPV vaccination pilot program was launched in Mirebalais, in central Haiti, supported by an initiative of Zanmi Lasante/Partners in Health (PIH). Secondary schools in the area were targeted with the goal of vaccinating 3,806 students. Zanmi Lasante reported that despite difficulties, including the earthquake of 2010, 86.8% of the targeted population received the second dose and 75.8% received the third dose of the HPV vaccine [21].

This study describes the results from a second HPV vaccination program in Haiti. In 2016, the Haitian Study Group on Kaposi's Sarcoma and Opportunistic Infections (GHESKIO) introduced the two-dose HPV vaccine in an urban setting in Haiti for the first time using two community-based vaccination strategies: a non-school-based approach and a school-based approach. The objective of this study was to assess the completion of the vaccination series among participants from both approaches, as well as to assess whether certain participant and guardian characteristics were associated with the completion of the vaccination series.

## Methods and materials

### HPV awareness and vaccination program

In 2016, GHESKIO, in collaboration with the MSPP and PIH, received 4,000 doses of the quadrivalent HPV vaccine from Merck Pharmaceuticals, effective against HPV 6, 11, and high-risk HPV 16 and 18.

In preparation for this vaccination effort, community health workers were trained to raise awareness of HPV, cervical cancer, modes of prevention, and the use of the HPV vaccine.; Awareness campaigns targeting adolescent girls were conducted at the GHESKIO clinic and throughout the neighboring communities and target schools from July 2016 through April 2017. As part of the campaign, meetings were organized in the community at large with school directors, parents, teachers, and students to educate them on HPV, cervical cancer, and the importance of the HPV vaccine.

From August 1, 2016 to April 5, 2017 participants of the vaccination program received up to two doses of GARDASIL® at a six-month interval, in accordance with WHO standards both then and now [22]. Participants were recruited via two community-based approaches: non-school-based and school-based. The non-school based group was vaccinated at the GHESKIO clinic in downtown Port-au-Prince and the school-based group was vaccinated at their respective schools.

Eligible participants were adolescent girls aged 9 to 14 who were willing to receive the vaccine and had parental consent. Participants with a history of allergy, asthma, or ongoing acute illness were excluded. The HIV status of participants was not an exclusion criterion.

For the non-school-based approach, eligible girls and their guardians were recruited from the GHESKIO adolescent and pediatric clinics. Guardians were invited to bring the adolescent girls to the GHESKIO clinic for vaccination. A participant's guardian was defined as the person who accompanied them to their first vaccination dose (e.g., family member, family friend, neighbor).

For the school-based approach, medical staff went to designated schools near the GHESKIO clinic to administer vaccines to eligible girls. Medical staff included nurses, community workers, and a data clerk. Thirteen schools were selected based on previous experiences working with the GHESKIO community team and the willingness of school directors to participate in the vaccination program. To receive the vaccine, recruited participants were required to have a signed permission sheet provided at school or sent to their home. The participant's guardian was defined as the individual who signed the permission sheet.

For both approaches, participants were assigned a field worker and given an appointment for their second dose during the first dose of the vaccination. For participants who missed their second dose appointment, the assigned field worker would conduct three phone calls, and if the participant could not be reached by phone, the assigned field worker would make a home visit. Additionally, girls who received the first dose of HPV vaccine and their families were invited to an end-of-year party held by GHESKIO, to maintain contact as an effort to ensure retention for the second dose of HPV vaccine.

## Data collection and analyses

A standardized questionnaire was administered to participants and their guardians in the non-school cohort and to participants only in the school cohort. The questionnaire was administered prior to vaccination, on the day of administration of each dose of the HPV vaccine. For the non-school cohort, the guardian was the person accompanying the participant to the vaccination. For the school cohort, the guardian was the person who signed the vaccination consent form; basic personal information and authorization to vaccinate were obtained via a letter sent out to participants' guardians, since they did not accompany the girls to school for the vaccination. The questionnaire was designed specifically for this study, and was administered in Creole, the native Haitian language. The study questionnaire can be found in S1 File. Side effects that occurred within 15 minutes of any vaccination dose were recorded in the questionnaire.

Comparisons were conducted between participants who received the first dose only and participants who received both doses using chi-square tests for categorical variables and Fisher's exact tests when patient count was fewer than 5 for any category. Wilcoxon rank-sum tests were used for continuous variables. Logistic regressions were used to identify characteristics associated with HPV vaccination series completion; covariates included age (continuous), education level (lower than secondary school, secondary school), menarche (yes, no), having been previously followed as patient in the GHESKIO clinic (yes, no), neighborhood distance from GHESKIO clinic (<1 kilometer, ≥1 kilometer; 1 kilometer was selected as a proxy for whether participants had ease of access to a GHESKIO health facility, measured from GHESKIO clinic to the central point of each neighborhood using the Google Maps ruler tool), guardian age (continuous), and guardian relationship to participant (mother, other). Given the low availability of information on participants' HIV status in the school cohort, it was not included in the regression model. Due to the low frequency of participants receiving only the first dose, the regression analysis was performed among all participants rather than by cohort, and cohort was included as a variable in the regression. P-values <0.05 were considered statistically significant. All statistical analyses were conducted using SAS version 9.4 (SAS Institute, Inc., Cary, North Carolina).

## Ethics statement

This project was approved by the GHESKIO Institutional Review Board (Comite des Droits Humains) on June 25[th], 2016. Parental consent was obtained prior to vaccination and documented in the study questionnaire; in the school cohort written parental consent was obtained, and in the non-school cohort parental consent was obtained verbally. Permission to conduct the project was obtained from the Ministry of Health (MSPP), who provided supervising staff to participate in this vaccination campaign.

## Results

### Vaccination consent and completion

A total of 2,445 adolescent girls participated in the awareness campaigns. 1,698 girls participated in the non-school cohort and 747 in the school cohort. The majority of participants in the awareness program consented and received the first dose of the HPV vaccine (non-school: 1,307 [77.0%], school: 687 [92.0%]). The vast majority of participants who had the first dose also received the second dose of the HPV vaccine in both the non-school (n = 1,199 [91.7%]) and school cohorts (n = 673 [98.0%]). Among all girls participating in the awareness program, 70.6% of girls in the non-school cohort and 90.1% of girls in the school cohort received both doses of the HPV vaccine. The mean time to the second vaccination was 5.8 months, ranging from 4.4 to 7.6 months. According to WHO guidelines, if the interval of time between dose 1 and 2 is less than 5 months, a third dose is recommended at least 6 months after dose 1. In our study, 0.2% (n = 5) participants received dose 2 within 5 months of receiving dose 1, none of which received a third dose. Overall, both the non-school and school cohorts completed the HPV vaccination series successfully.

### Characteristics of adolescents with ≥1 dose of the HPV vaccine

Overall, the mean age of participants at the first vaccination was 11.7 years old (standard deviation [SD] = 1.5 years) and more than half had not experienced menarche at the time of the first vaccination (65.3%, Table 1); this was similar in the school and non-school cohorts. Participants previously followed as patients in the GHESKIO clinic represented 30.6% and 36.7%

**Table 1. Characteristics of participating adolescents with at least 1 dose of HPV vaccine and their guardians, overall and by cohort.**

| | All Participants, N(%) | | | | Non-School, N(%) | | | | School, N(%) | | | |
|---|---|---|---|---|---|---|---|---|---|---|---|---|
| | Total | Received 1st dose only | Received both doses | | Total | Received 1st dose only | Received both doses | | Total | Received 1st dose only | Received both doses | |
| Parameter | (N = 1,994) | (N = 122) | (N = 1,872) | P-value[a] | (N = 1,307) | (N = 108) | (N = 1,199) | P-value[a] | (N = 687) | (N = 14) | (N = 673) | P-value[a] |
| *Participant Characteristics* | | | | | | | | | | | | |
| Age, mean (SD) | 11.7 (1.5) | 11.7 (1.5) | 11.7 (1.5) | 0.714 | 11.8 (1.4) | 11.7 (1.4) | 11.8 (1.4) | 0.424 | 11.5 (1.6) | 11.4 (1.7) | 11.5 (1.6) | 0.691 |
| Previous GHESKIO clinic patient, n (%)[b] | 541 (27.1) | 24 (19.7) | 517 (27.6) | 0.055 | 470 (36.0) | 24 (22.2) | 446 (37.2) | 0.002* | 71 (10.3) | 0 (0.0) | 71 (10.5) | 0.382 |
| Experienced menarche, n (%) | 690 (34.6) | 32 (26.2) | 658 (35.1) | 0.044* | 480 (36.7) | 30 (27.8) | 450 (37.5) | 0.044* | 210 (30.6) | 2 (14.3) | 208 (30.9) | 0.247 |
| HIV data available, n (%) | 563 (28.2) | 34 (27.9) | 529 (28.3) | | 524 (40.1) | 33 (30.6) | 491 (41.0) | | 39 (5.7) | 1 (7.1) | 38 (5.6) | |
| HIV positive, n (%)[c] | 38 (6.7) | 4 (3.3) | 34 (1.8) | 0.276 | 38 (7.3) | 4 (12.1) | 34 (6.9) | 0.287 | 0 (0.0) | 0 (0.0) | 0 (0.0) | – |
| Education data available, n (%) | 1,961 (98.3) | 118 (96.7) | 1,843 (98.5) | | 1,274 (97.5) | 104 (96.3) | 1,170 (97.6) | | 687 (100) | 14 (100) | 673 (100) | |
| Secondary school, n (%)[c] | 398 (20.3) | 17 (13.9) | 381 (20.4) | 0.101 | 205 (16.1) | 14 (13.5) | 191 (16.3) | 0.446 | 193 (28.1) | 3 (21.4) | 190 (28.2) | 0.767 |
| Neighborhood <1km from GHESKIO clinic, n (%)[d] | 916 (45.9) | 67 (54.9) | 849 (45.4) | 0.040* | 769 (58.8) | 65 (60.2) | 704 (58.7) | 0.766 | 147 (21.4) | 2 (14.3) | 145 (21.5) | 0.745 |
| *Guardian Characteristics* | | | | | | | | | | | | |
| Age | | | | | | | | | | | | |
| Mean (SD) | 35.4 (10.3) | 33.6 (10.1) | 35.5 (10.3) | 0.017* | 33.7 (11.0) | 33.3 (10.3) | 33.8 (11.0) | 0.614 | 38.5 (8.0) | 35.9 (7.7) | 38.6 (8.0) | 0.157 |
| Min-Max [Q1, Q3] | (18–77) [28, 41] | (18–63) [27, 40] | (18–77) [29, 42] | | (18–77) [25, 40] | (18–63) [27, 39] | (18–77) [25, 40] | | (19–71) [33, 43] | (25–54) [31, 42] | (19–71) [33, 43] | |
| Sex, female, n (%) | 1,757 (88.1) | 100 (82.0) | 1,657 (88.5) | 0.030* | 1,093 (83.6) | 87 (80.6) | 1,006 (83.9) | 0.368 | 664 (96.7) | 13 (92.9) | 651 (96.7) | 0.382 |
| Relationship to girl, mother, n (%) | 1,075 (53.9) | 40 (32.8) | 1,035 (55.3) | <0.001* | 439 (33.6) | 27 (25.0) | 412 (34.4) | 0.049* | 636 (92.6) | 13 (92.9) | 623 (92.6) | 1.000 |

Abbreviations: GHESKIO, Haitian Study Group on Kaposi's Sarcoma and Opportunistic Infections; HIV, human immunodeficiency virus; HPV, human papillomavirus; Max, maximum; Min, minimum; N, number; Q1: first quartile; Q3: third quartile; SD, standard deviation. Notes: * *P*< 0.05 was considered statistically significant.

[a] *P*-values were calculated for the comparison between girls who received the 1st dose only and girls who received both doses using Wilcoxon rank-sum tests for continuous variables and chi-square for categorical variables. Fisher's exact test were used when a count was <5.

[b] Indicated that the participant has been previously followed as patient in the GHESKIO clinic.

[c] Proportions are presented among those with data available.

[d] Distance from the neighborhood where the girl and guardian were living relative to GHESKIO clinic.

of the school and non-school cohorts, respectively. HIV status was not available for most participants (71.8%), especially for the school cohort (unavailable for 94.3%), but prevalence of HIV was low (6.7%) among those with data available. More than half (58.8%) of the non-school cohort were from neighborhoods within 1 kilometer of the GHESKIO clinic, whereas only around one fifth (21.4%) were from neighborhoods within 1 kilometer of the GHESKIO clinic in the school cohort.

The mean age of guardians was 35.4 years old (SD = 10.3 years) and the vast majority (88.1%) of participants had female guardians. In the non-school cohort, a third of the guardians were the participants' mother (33.6%), followed by extended family (26.7%) and companions other than father, older sibling, or godparents (25.6%). Most of these other companions were friends or neighbors of the participants' parents. In the school cohort, almost all guardians were the participants' mother (92.6%).

## Vaccination side effects

A total of 50 participants (2.5%) who received the first dose and 157 participants (8.4%) who received the second dose experienced malaise within two days. Among the non-school cohort,

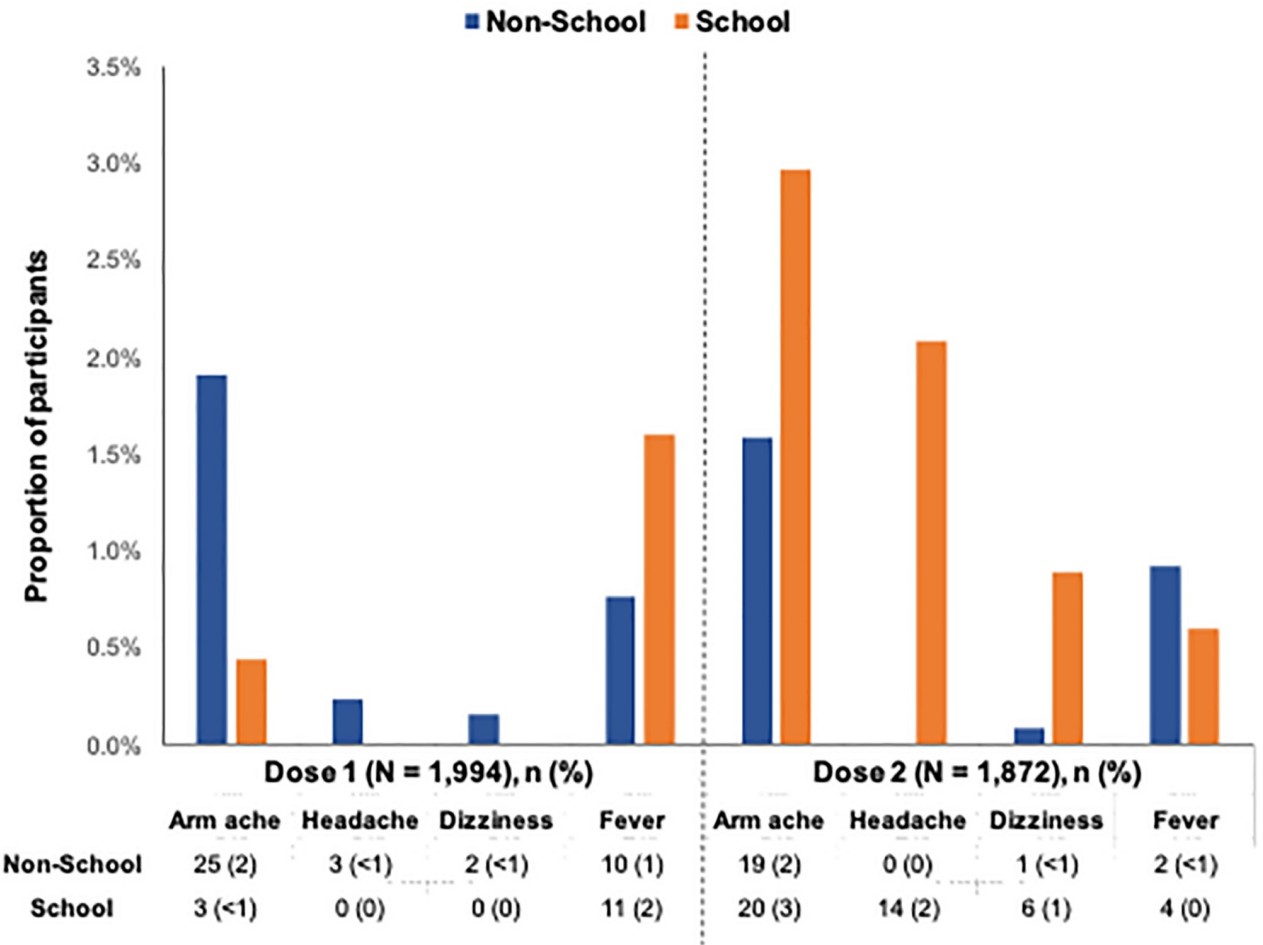

**Fig 1. Side effects within two days among participants with at least 1 dose of the HPV vaccine by cohort.** [a,b] a Only 55 participants (non-school:41, school: 14) and 66 participants (non-school: 22, school: 44) reported specific side effects at dose 1 and dose 2, respectively. However, a total of 50 and 157 participants reported malaise within 2 days at dose 1 and dose 2, respectively. b A non-prespecified event reported by one participant in the non-school cohort was allergic reaction after vaccine.

the most commonly reported side effect within two days was arm ache at both dose 1 (1.9%; n = 25) and dose 2 (1.6%; n = 19) (Fig 1). For the school cohort, the most common side effects were fever at dose 1 (1.6%; n = 11), and arm ache (3.0%; n = 20) and headache (2.1%; n = 14) at dose 2.

## Characteristics associated with vaccination series completion

In univariate assessments among all participants who received the first HPV vaccine dose, a higher percentages of girls who received both doses lived less than 1 kilometer from the GHES-KIO clinic (54.9 v. 45.4%; $P$ = 0.040, Table 1), and had experienced menarche (35.1% v. 26.2%; $P$ = 0.044), compared to girls who only received one dose. Receiving both doses was associated with having an older Guardian (35.5 v. 33.6 years old; $P$ = 0.017) who is female (88.5% v. 82.0%; $P$ = 0.030) and the girl's mother (55.3% v. 32.8%; $P$<0.001).

Overall, the proportion of participants who were patients at GHESKIO among those who completed the vaccination series was not quite statistically different from that of participants who have not been a patient at the clinic ($P$ = 0.055; Table 1). However, in the non-school

cohort, 37% of patients who completed the HPV vaccine series were previously followed at a GHESKIO clinic versus only 22% among patients who did not complete the series ($P<0.002$). In the non-school cohort, among patients who received the first dose and had never been a patient at a GHESKIO clinic (n = 837), 753 (90.0%) completed the vaccination series. For this subset of girls, participant and guardian characteristics were not significantly different between those who completed the vaccine series and those who did not.

After adjusting for participant age, education, whether the participant was a GHESKIO patient or not, cohort (school or non-school), neighborhood distance from GHESKIO clinic, and guardian characteristics, girls who had experienced menarche had 1.87 times higher odds of completing the vaccination series than those who had not experienced menarche (95% confidence interval [CI], 1.11–3.14; $P$ = 0.019; Table 2). Participants previously followed as a patient at the GHESKIO clinic had 2.17 times the odds of receiving a second dose than participants who were not (95% CI, 1.32–3.58; $P$ = 0.002). Additionally, participants who received the vaccine in a school-based setting had 4.17 times higher odds of vaccination series completion than those who were vaccinated in a non-school-based setting (95% CI, 2.14–8.12; $P<0.001$). Participants with guardians aged 26 to 35 years had lower odds of completing the vaccination series, compared to those with guardians aged 46 or above (odds ratio 0.64, 95% CI, 0.36–1.15, $P$ = 0.006). Participant age, education, living within 1 kilometer of GHESKIO clinic, and guardian relationship were not significantly associated with vaccination series completion in multivariable regressions.

## Discussion

This study demonstrates that both non-school and school-based approaches to HPV vaccination in an urban setting in Haiti resulted in high rate of 2-dose HPV vaccination series completion. Among all girls who participated in the awareness program in the non-school and school cohort, 77.0% and 92.0% received a first dose of the vaccine, and 70.6% and 90.1% received both doses of the HPV vaccine, respectively.

The high consent rate (92.0%) and vaccination series completion rate among girls who participated in the awareness program (90.1%) in the school-based setting of this vaccination program are consistent with findings from other studies assessing a school-based approach in resource-limited countries [23]. In Rwanda, media campaigns followed by a national HPV vaccination program in 2011 targeting girls in the sixth grade resulted in 88,927 of the 94,141 (94%) eligible girls in school in the country receiving the three-dose vaccination series in a school-based setting [15, 16]. In a national school-based HPV vaccination program in South Africa, 91% of schools were reached with vaccination sessions, and 86.6% of fourth grade girls more than 8 years old received the two-dose vaccination series [19]. The high adherence between doses observed in our program (98.0%) is also consistent with that observed in Partners In Health/Zanmi Lasante's 2009 school-based program in rural communities (75.8%) [21].

Participants vaccinated in the non-school-based setting were still likely to receive both doses of the vaccine: 70.6% of the girls who participated in the awareness program at the health center also received the second dose of the vaccine. This is also supported by findings in resource-limited regions. As part of the 2011 national HPV vaccination program in Rwanda discussed above, the health facility-based program, which targeted sixth-grade girls absent from school on vaccination days and girls of eligible age but were not enrolled in school, vaccinated 85.2% of all eligible girls nationally [15]. In Mexico, the roll out of HPV vaccines in 2008 and delivered through mobile health clinics for girls aged 12 to 16 years in targeted

**Table 2. Multivariable regression of completion of the vaccination series among participating adolescents who received at least 1 dose of HPV vaccination.[a]**

| | All participants (N = 1,960) | | |
|---|---|---|---|
| Variable | Adjusted OR | 95% CI | P-Value |
| *Participant characteristics* | | | |
| Participant age (years) | 0.92 | (0.78, 1.08) | 0.311 |
| Participant education | | | |
| Secondary | 1.10 | (0.62, 1.97) | 0.744 |
| <Secondary | | Reference | |
| Experienced menarche | | | |
| Yes | 1.87 | (1.11, 3.14) | 0.019* |
| No | | Reference | |
| Patient at GHESKIO clinic[b] | | | |
| Yes | 2.17 | (1.32, 3.58) | 0.002* |
| No | | Reference | |
| Cohort | | | |
| School | 4.17 | (2.14, 8.12) | <0.001* |
| Non-school | | Reference | |
| Neighborhood distance from GHESKIO clinic[c] | | | |
| <1km | 1.05 | (0.71, 1.57) | 0.800 |
| ≥1km | | Reference | |
| *Guardian characteristics* | | | |
| Guardian age (years) | | | |
| 18 to 25 | 1.23 | (0.64, 2.36) | 0.220 |
| 26 to 35 | 0.64 | (0.36, 1.15) | 0.006* |
| 36 to 45 | 1.13 | (0.59, 2.16) | 0.419 |
| 46 or older | | Reference | |
| Guardian relationship | | | |
| Mother | 1.50 | (0.93, 2.42) | 0.098 |
| Other[d] | | Reference | |

Abbreviations: CI, confidence interval; GHESKIO, Haitian Study Group on Kaposi's Sarcoma and Opportunistic Infections; OR, odds ratio. Note:

* $P<0.05$ were considered statistically significant.

[a] Multivariable regression analysis include all the variables listed in the table as independent variables, and completion of the HPV vaccination series as the dependent variable. Patients with missing information education level, menarche, GHESKIO patient history, neighborhood, or guardian information (1.7%; n = 34) were excluded from the regression.

[b] Indicated that the participant has been previously followed as patient in the GHESKIO clinic.

[c] Distance from the neighborhood where the girl and guardian were living relative to GHESKIO clinic.

[d] Other guardian relationships included other family members (father, aunt/uncle, cousin, sibling, grandparent, parent/sibling-in-law, unspecified family member), godparents, friends of family, neighbor, and caregiver ("akonpayatè").

municipalities with low human development index achieved 98% and 81% of coverage of the first and third dose of the vaccine, respectively [20].

This study also shows that that after the initial consent and first dose of vaccination, patient attrition was low, with successful completion of the HPV vaccination series observed in 91.7% and 98.0% in the non-school and school cohort, respectively. The results suggest that reaching the targeted adolescents and obtaining initial consent may be the largest obstacles to adolescent girls receiving an initial HPV vaccination in the context of urban Haiti but that high retention of girls for the second dose can be feasibly achieved in both school and non-school settings.

As previous research has shown, a successful expansion of an HPV vaccination campaign to a national-level in a low-to-middle-income country requires local and international support and financial investment [24–26]. A key aspect highlighted by similar programs in Mozambique was that out-of-school and unenrolled girls would be missed by school-based vaccination programs; an adaptable model that would allow communities to target this demographic via both school-based and non-school based vaccination programs is recommended. The high rate of successful completion in our study in both the non-school and school cohorts, provides evidence that such adaptable models would be appropriate in Haiti.

This study is subject to some limitations. The vaccination campaign was conducted only in the metropolitan area of Port-au Prince and findings may not be generalizable to all of Haiti. Additionally, this study did not record information for participants who did not consent to the vaccine after participating in the awareness program, or reasons for non-completion of vaccine series, which would be useful information to capture in future studies. This study also only included patients who voluntarily agreed to participate in the awareness program, who may have already had an interest in the vaccine, and may have behaved different from a more general population of adolescents who are not willing to learn about HPV and vaccination. Further, the multivariable regressions could be biased due to unmeasured confounding from unavailable variables, such as HIV status, socioeconomic status, and guardian education level. Finally, this study did not recruit adolescent boys to receive the HPV vaccine. Any long-term HPV vaccination program will have to account for the vaccination of adolescent boys to prevent HPV-related cancers.

## Public health implications

Since 2006, safe and effective vaccines against HPV have been available, yet by 2014, only 0.1% and 1.0% of all female aged 10–20 years were estimated to have received the full course of HPV vaccination in low-middle-income and low-income countries, respectively [27, 28]. Thus, the populations with the highest HPV incidence and mortality remain unvaccinated, and there is a need for rapid roll-out of the vaccine to these regions.

Studies on HPV vaccination programs in low- and middle-income countries show both school-based and health facility-based strategies achieved high overall success measured by vaccine uptake and adherence between doses [5, 23]. The success of different types of strategies, including both school-based and health-facility-based approaches, supports that there is no "gold standard" for designing an HPV vaccination program [23]. The high vaccination coverage and adherence between first and subsequent vaccine doses across the assessed studies suggest that approaches implemented by vaccination programs should be tailored to the challenges and needs of the region in which it is implemented.

Obstacles identified by other HPV vaccination programs in low- and middle-income countries include reaching and maintaining follow-up [5]. The high adherence of our program shows that initial recruitment is feasible in both school and non-school settings and that series completion is very high among girls who initiate the series. Future efforts are warranted for the expansion of the reach and education of girls and guardians about HPV vaccination and cervical cancer, especially outside of a school setting.

This study is the first of its kind in an urban Haitian setting, and the results support that a large-scale HPV vaccination program in urban Haiti would be well-received. It demonstrates the successful completion of the two-dose HPV vaccination series in both school-based and non-school-based settings, with over 90% of patients completing the vaccination series in both settings conditional on consent and administration of the first dose. The findings support the future feasibility of similar HPV vaccination programs in Haiti.

## Supporting information

**S1 File. HPV vaccination questionnaire administered by GHESKIO (Creole) for school-based and non-school based strategies.**
(PDF)

**S1 Dataset.**
(XLSX)

## Acknowledgments

We would like to acknowledge the GHESKIO and MSPP staff who participated in this project on a voluntary basis, Partners in Health for providing guidance and insight, Merck Pharmaceuticals for providing the vaccine, and our colleagues for providing feedback on the manuscript and analyses: Erin Cook, Henry Ulmer, and Pierre Cremieux (Analysis Group, Inc.).

## Author Contributions

**Conceptualization:** Cynthia Riviere, Tatiana Bell, Yonie Cadot, Christian Perodin, Benedict Charles, Claudin Bertil, Jean William Pape, Marie Marcelle Deschamps.

**Data curation:** Cynthia Riviere, Tatiana Bell, Yonie Cadot, Christian Perodin, Benedict Charles, Claudin Bertil, Jean William Pape, Marie Marcelle Deschamps.

**Formal analysis:** Jazreel Cheung, Shalmali Bane, Hoi Ching Cheung, Marie Marcelle Deschamps.

**Investigation:** Cynthia Riviere, Tatiana Bell, Yonie Cadot, Benedict Charles, Claudin Bertil, Jean William Pape, Marie Marcelle Deschamps.

**Methodology:** Cynthia Riviere, Tatiana Bell, Yonie Cadot, Christian Perodin, Benedict Charles, Claudin Bertil, Jean William Pape, Marie Marcelle Deschamps.

**Project administration:** Cynthia Riviere, Tatiana Bell, Christian Perodin, Benedict Charles, Claudin Bertil, Marie Marcelle Deschamps.

**Resources:** Cynthia Riviere.

**Supervision:** Cynthia Riviere, Tatiana Bell, Yonie Cadot, Christian Perodin, Benedict Charles, Claudin Bertil, Jean William Pape, Marie Marcelle Deschamps.

**Writing – original draft:** Jazreel Cheung, Shalmali Bane, Hoi Ching Cheung.

**Writing – review & editing:** Cynthia Riviere, Tatiana Bell, Yonie Cadot, Jazreel Cheung, Shalmali Bane, Hoi Ching Cheung, Jean William Pape, Marie Marcelle Deschamps.

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
