## [Decision Letter · Decision Letter 0]

26 Jan 2021

PONE-D-20-36531

Success of Community Approach to HPV Vaccination in School-Based and Non-School-Based Settings in Haiti

PLOS ONE

Dear Dr. Bane,

Thank you for submitting your manuscript to PLOS ONE. After careful consideration, we feel that it has merit but does not fully meet PLOS ONE’s publication criteria as it currently stands. Therefore, we invite you to submit a revised version of the manuscript that addresses the points raised during the review process.

We look forward to receiving your revised manuscript.

Kind regards,

Orvalho Augusto, MD, MPH

Academic Editor

PLOS ONE

Additional Editor Comments:

This is a relevant work for Haiti and potentially for other Caribbean countries. This is a short report for a short community HPV vaccination campaign in Urban Haiti for adolescent girls.

It had a high degree of success given the rates they managed to reach.

Few questions minor issues:

1. For descriptive analysis on table 1, please add more descriptives for age of the guardian. Quartiles and range would be useful.

2. Explain how the distance were measured. How this 1km cut-off was established.

3. How the adverse effects were collected. I do not see a description of this in the manuscript.

4. For the logistic regression:

- Good that all covariates were introduced. Is there any reason besides availability for these small set of covariates?

- Why the age of guardian in the regression was introduced as linear?

- Did you try more levels of distance?

5. In a supplementary (not necessary for the main manuscript) would be good to have a more detailed list the guardian relationship

Journal Requirements:

3. During our internal checks, the in-house editorial staff noted that you conducted research or obtained samples in another country. Please check the relevant national regulations and laws applying to foreign researchers and state whether you obtained the required permits and approvals. Please address this in your ethics statement in both the manuscript and submission information.  In addition, please ensure that you have suitably acknowledged the contributions of any local collaborators involved in this work in your authorship list and/or Acknowledgements. Authorship criteria is based on the International Committee of Medical Journal Editors (ICMJE) Uniform Requirements for Manuscripts Submitted to Biomedical Journals - for further information please see here: https://journals.plos.org/plosone/s/authorship.

We note that one or more of the authors are employed by a commercial company: Analysis Group, Inc.

4.1. Please provide an amended Funding Statement declaring this commercial affiliation, as well as a statement regarding the Role of Funders in your study. If the funding organization did not play a role in the study design, data collection and analysis, decision to publish, or preparation of the manuscript and only provided financial support in the form of authors' salaries and/or research materials, please review your statements relating to the author contributions, and ensure you have specifically and accurately indicated the role(s) that these authors had in your study. You can update author roles in the Author Contributions section of the online submission form.

4.2. Please also provide an updated Competing Interests Statement declaring this commercial affiliation along with any other relevant declarations relating to employment, consultancy, patents, products in development, or marketed products, etc.  

Reviewers' comments:

Reviewer's Responses to Questions

**Comments to the Author**

1. Is the manuscript technically sound, and do the data support the conclusions?

Reviewer #1: Yes

2. Has the statistical analysis been performed appropriately and rigorously? 

Reviewer #1: Yes

3. Have the authors made all data underlying the findings in their manuscript fully available?

Reviewer #1: Yes

4. Is the manuscript presented in an intelligible fashion and written in standard English?

Reviewer #1: Yes

5. Review Comments to the Author

Reviewer #1: In this paper the Authors performed an assessment of two modalities of delivering HPV vaccines to adolescent girls in an urban setting in Port-au-Prince Haiti. In particular, they estimated descriptive statistics and utilized logistic regression models to examine characteristics associated with completion of vaccination doses for each delivery method. This study is very interesting however 5 points need to be addressed:

1. Abstract page 2. Methods. Change this section to read “Data collection was conducted during HPV vaccination campaigns in Port -au -Prince between August 2016 and April 2017. Descriptive statistics and logistic regression models were used to examine characteristics associated with vaccination series completion of school based and non-school based vaccination delivery modalities.”

2. Abstract page 2. Conclusions. Rewrite to read "Vaccination in school- and non-school-based settings was successful, suggesting that a nationwide HPV vaccination campaign using either approach would be successful".

3. Main text Methods and materials: Page 5 Line 77. End the sentence at after HPV vaccine. The next line 78 should then read “Awareness campaigns targeting adolescent girls were conducted at the GHESKIO clinic and throughout neighboring communities and target schools from July 2016 through April 2017”

4. Literature review is not enough. There are some articles, which must be added to literature review: a) Soi, C. Human papillomavirus vaccine delivery in Mozambique: identification of implementation performance drivers using the Consolidated Framework for Implementation Research (CFIR); b) Soi, C. Implementation Strategy and Cost of Mozambique’s HPV vaccine demonstration project; c) Soi, C. Global health systems partnerships: A mixed methods analysis of HPV vaccine delivery network actors in Mozambique.

5. DOI of all the references must be added (you can use "" ext-link-type="uri" xlink:type="simple">https://crossref.org/").

6. PLOS authors have the option to publish the peer review history of their article (what does this mean?). If published, this will include your full peer review and any attached files.

Reviewer #1: No

---

## [Author Response · Author response to Decision Letter 0]

24 Mar 2021

Editor’s Comments

This is a relevant work for Haiti and potentially for other Caribbean countries. This is a short report for a short community HPV vaccination campaign in Urban Haiti for adolescent girls.

It had a high degree of success given the rates they managed to reach.

Comment 1: For descriptive analysis on table 1, please add more descriptives for age of the guardian. Quartiles and range would be useful.

Response to Comment 1: We thank the Editor for this comment and have added the following information to Table 1, Pg. 10 for guardian age: minimum, maximum, 1st quartile, median, 3rd quartile. 

Comment 2: Explain how the distance were measured. How this 1km cut-off was established.

Response to Comment 2: The 1km cut-off was selected a priori to data analysis as a proxy for whether participants had ease of access to a GHESKIO health facility. Distance was measured using the Google Maps ruler tool from GHESKIO clinic to the central point of each neighborhood. This information has also been added to the manuscript on Pg. 8: 

“Logistic regressions were used to identify characteristics associated with HPV vaccination series completion; covariates included age (continuous), education level (lower than secondary school, secondary school), menarche (yes, no), having been previously followed as patient in the GHESKIO clinic (yes, no), neighborhood distance from GHESKIO clinic (1 kilometer, ≥1 kilometer; 1 kilometer was selected as a proxy for whether participants had ease of access to a GHESKIO health facility, measured from GHESKIO clinic to the central point of each neighborhood using the Google Maps ruler tool), guardian age (continuous), and guardian relationship to participant (mother, other).”

Comment 3: How the adverse effects were collected. I do not see a description of this in the manuscript.

Response to Comment 3: We thank the Editor for their comment and agree that clarity with regards to how adverse effects were collected is needed. Adverse effects were collected within 15 minutes of each vaccination dose and recorded on that participants questionnaire. This information has also been added to the manuscript on Pg. 8: 

“Side effects that occurred within 15 minutes of any vaccination does were recorded in the questionnaire.”

Comment 4: For the logistic regression: Good that all covariates were introduced. Is there any reason besides availability for these small set of covariates? Why the age of guardian in the regression was introduced as linear? Did you try more levels of distance?

Response to Comment 4: We thank the Editor for these suggestions. Covariates included in this analysis were constrained by data availability.

In response to the feedback on guardian age, we modeled guardian age as a categorical variable with four levels: 18 to 25 years, 26 to 35 years, 36 to 45 years, 46 or older. Girls with guardians aged 26 to 35 years had statistically significantly lower odds of receiving at least 1 dose. We have updated Table 2 to present the results of the regression using categorical instead of continuous guardian age as covariate (see Pg. 14 of the manuscript).

We did not initially attempt to stratify by additional levels of distance. A sensitivity analysis was performed to include 3 levels of distance: 1 km, 1 to 5 km, and 5 km from GHESKIO clinic. Odds ratios and significance of covariates are largely consistent with the original analysis, and no statistically significant difference between the three distance levels was observed. Hence, we did not change our analysis to add additional levels of distance, and have presented the sensitivity findings as reported in revised paper, and the attached copy of responses to reviewers (not included here due to table formatting constraints). 

Comment 5: In a supplementary (not necessary for the main manuscript) would be good to have a more detailed list the guardian relationship

Response to Comment 5: Detailed list of guardian relationships has been included in the manuscript as a footnote to Table 2, on Pg. 14. The complete list of guardian relationships in the sample are as follows:

Relationship Count (%)

Mother/Father (“Manman/Papa”) 1,162 (58.3%)

Neighbor (“Vwazin/Vwazine”) 310 (15.6%)

Aunt/Uncle (“Matant/Mononk”) 163 (8.2%)

Cousin (“Gran konzin/Gran kouzen”) 152 (7.6%)

Older sister/Older brother (“Gran sè/Gran frè”) 104 (5.2%)

Grandmother/Grandfather (“Granmè/Granpè”) 34 (1.7%)

Godmother/Godfather (“Marenn/Parenn”) 16 (0.8%)

Mother-in-law/father-in-law (“Bèl mè /Bo pè”) 15 (0.8%)

Mother friend (“Zanmi manman”) 14 (0.7%)

Other relation (“Lòt relasyon”), please specify 24 (1.2%)

Caregiver (“Akonpayatè”) 13 (0.7%)

Brother-in-law/Sister-in-law (“Bel sè / Bo frè”) 4 (0.2%)

Godmother (“Makomè”) 3 (0.2%)

Godson/Goddaughter (“Fiyel”) 2 (0.1%)

Nephew/Niece (“Neve”) 1 (0.1%)

Family (“Fanmi”) 1 (0.1%)

 

Reviewer #1 Comments

In this paper the Authors performed an assessment of two modalities of delivering HPV vaccines to adolescent girls in an urban setting in Port-au-Prince Haiti. In particular, they estimated descriptive statistics and utilized logistic regression models to examine characteristics associated with completion of vaccination doses for each delivery method. This study is very interesting however 5 points need to be addressed:

Comment 1: Abstract page 2. Methods. Change this section to read “Data collection was conducted during HPV vaccination campaigns in Port -au -Prince between August 2016 and April 2017. Descriptive statistics and logistic regression models were used to examine characteristics associated with vaccination series completion of school based and non-school based vaccination delivery modalities.”

Response to Comment 1: We thank the reviewer for their comment and have incorporated the suggested language in the abstract on Pg. 2. 

Comment 2: Abstract page 2. Conclusions. Rewrite to read "Vaccination in school- and non-school-based settings was successful, suggesting that a nationwide HPV vaccination campaign using either approach would be successful".

Response to Comment 2: See response to Comment 1. 

Comment 3: Main text Methods and materials: Page 5 Line 77. End the sentence at after HPV vaccine. The next line 78 should then read “Awareness campaigns targeting adolescent girls were conducted at the GHESKIO clinic and throughout neighboring communities and target schools from July 2016 through April 2017”

Response to Comment 3: We have incorporated the suggested language on Pg. 5. 

Comment 4: Literature review is not enough. There are some articles, which must be added to literature review: a) Soi, C. Human papillomavirus vaccine delivery in Mozambique: identification of implementation performance drivers using the Consolidated Framework for Implementation Research (CFIR); b) Soi, C. Implementation Strategy and Cost of Mozambique’s HPV vaccine demonstration project; c) Soi, C. Global health systems partnerships: A mixed methods analysis of HPV vaccine delivery network actors in Mozambique.

Response to Comment 4: The suggested references have been incorporated into the discussion section on Pg. 17:

“As previous research has shown, a successful expansion of an HPV vaccination campaign to a national-level in a low-to-middle-income country requires local and international support and financial investment [24-26]. A key aspect highlighted by similar programs in Mozambique was that out-of-school and unenrolled girls would be missed by school-based vaccination programs; an adaptable model that would allow communities to target this demographic via both school-based and non-school based vaccination programs is recommended. The high rate of successful completion in our study in both the non-school and school cohorts, provides evidence that such adaptable models would be appropriate in Haiti.”

Comment 5: DOI of all the references must be added (you can use "https://crossref.org/").

Response to Comment 5: DOI inputs for all references have been updated.

---

## [Decision Letter · Decision Letter 1]

14 Apr 2021

PONE-D-20-36531R1

Success of Community Approach to HPV Vaccination in School-Based and Non-School-Based Settings in Haiti

PLOS ONE

Dear Dr. Bane,

Thank you for submitting your manuscript to PLOS ONE. After careful consideration, we feel that it has merit but does not fully meet PLOS ONE’s publication criteria as it currently stands. Therefore, we invite you to submit a revised version of the manuscript that addresses the points raised during the review process.

If applicable, we recommend that you deposit your laboratory protocols in protocols.io to enhance the reproducibility of your results. Protocols.io assigns your protocol its own identifier (DOI) so that it can be cited independently in the future. For instructions see: http://journals.plos.org/plosone/s/submission-guidelines#loc-laboratory-protocols. Additionally, PLOS ONE offers an option for publishing peer-reviewed Lab Protocol articles, which describe protocols hosted on protocols.io. Read more information on sharing protocols at https://plos.org/protocols?utm_medium=editorial-emailutm_source=authorlettersutm_campaign=protocols.

We look forward to receiving your revised manuscript.

Kind regards,

Orvalho Augusto, MD, MPH

Academic Editor

PLOS ONE

Journal Requirements:

Additional Editor Comments (if provided):

New issues came up from a second reviewer. Please see below.

Reviewers' comments:

Reviewer's Responses to Questions

**Comments to the Author**

1. If the authors have adequately addressed your comments raised in a previous round of review and you feel that this manuscript is now acceptable for publication, you may indicate that here to bypass the “Comments to the Author” section, enter your conflict of interest statement in the “Confidential to Editor” section, and submit your "Accept" recommendation.

Reviewer #1: All comments have been addressed

Reviewer #2: (No Response)

2. Is the manuscript technically sound, and do the data support the conclusions?

Reviewer #1: Yes

Reviewer #2: Partly

3. Has the statistical analysis been performed appropriately and rigorously? 

Reviewer #1: Yes

Reviewer #2: Yes

4. Have the authors made all data underlying the findings in their manuscript fully available?

Reviewer #1: Yes

Reviewer #2: No

5. Is the manuscript presented in an intelligible fashion and written in standard English?

Reviewer #1: Yes

Reviewer #2: Yes

6. Review Comments to the Author

Reviewer #1: The authors have responded to all the reviewer comments. The manuscript describes a technically sound piece of scientific research with data that supports the conclusions. Experiments have been conducted rigorously, with appropriate controls, replication, and sample sizes. The conclusions are drawn appropriately based on the data presented. All data underlying the findings described in the manuscript are fully available without restriction.

Reviewer #2: I have some major concerns.

I understood the study participants included patients with Kaposi sarcoma and healthy adolescent girls. This is not clear in the abstract , introduction and methodology.

In the Line No 219-221 authors state that Overall, the proportion of participants who were GHESKIO patients among those who completed the vaccination series was not quite statistically different from that of participants who have not been a patient at the clinic.

Article need more clarity.

Abstract

Objectives Line No 24-26 Specify the age of study participants

Results Line 34 The term "Patient" is confusing, As I understood the study included both patients as well as healthy adolecscents term patient to be replaced with vaccine recipients or vaccinees.

Instead of patient history-medical/surgical history of study participents will be appropriate.

Line No 72 GHESKIO -Is the vaccine introduced in only patients with Kaposi sarcoma. Mention other healthy participants.

Line 81 Abbreviation for PIH

Line 102 Methods and materials

Define PAtient /study participants

Line 178 whereas only around one fifth (21.4%) were pregnant at the time of

vaccination, all of whom received both doses of the vaccine. HPV vaccine is not administered in pregnant women.

Justify this.

Line 228 After adjusting for participant age, education, GHESKIO patient status-Authors have to define GHESKIO patient status

7. PLOS authors have the option to publish the peer review history of their article (what does this mean?). If published, this will include your full peer review and any attached files.

Reviewer #1: No

Reviewer #2: No

---

## [Author Response · Author response to Decision Letter 1]

7 May 2021

Reviewer #1 Comments

Comment 1: I have some major concerns. I understood the study participants included patients with Kaposi sarcoma and healthy adolescent girls. This is not clear in the abstract , introduction and methodology. 

Response 1: We thank the reviewer for this comment and the opportunity to make this clear in our manuscript. GHESKIO (the Haitian Study Group on Kaposi’s Sarcoma and Opportunistic Infections) is the community organization that conducted this study in Port-au-Prince. GHESKIO was founded in 1982 and was the world’s first institution dedicated to fighting HIV/AIDS. Today, GHESKIO provided HIV testing and treatment, maternal-child health and nutrition services, treatment for sexually transmitted infections, and many other social services. 

In this study, the population included adolescent girls aged 9 to 14 who were willing to receive the vaccine and had parental consent (lines 97-98); the study population did not include individuals with Kaposi’s Sarcoma. In order to make this clear to readers, we have edited language in the manuscript from “GHESKIO patient” to “patient at a GHESKIO clinic” where applicable (Table 1, line 219, 222-223, 225, 228-230, 239, Table 2).

Comment 2: In the Line No 219-221 authors state that “Overall, the proportion of participants who were GHESKIO patients among those who completed the vaccination series was not quite statistically different from that of participants who have not been a patient at the clinic.” Article need more clarity.

Response 2: See Response 1. This statement refers to whether the participant was a former GHESKIO clinic patient or not. We have added a reference to Table 1 for clarity (lines 219-221). 

Comment 3: Abstract Objectives Line No 24-26 Specify the age of study participants

Response 3: The abstract has been updated accordingly. See below (lines 24-26):

“Objectives. To assess the success of a human papillomavirus (HPV) vaccination program among adolescent girls aged 9-14 years in Haiti and to understand predictors of completion of a two-dose HPV vaccination series.”

Comment 4: Results Line 34 The term "Patient" is confusing, As I understood the study included both patients as well as healthy adolescents term patient to be replaced with vaccine recipients or vaccinees. Instead of patient history-medical/surgical history of study participants will be appropriate.

Response 4: See response 1. Additionally, we have updated the language to make clear that we are referring to whether the participant was a former patient at a GHESKIO clinic. See below (lines 34-35): 

“Menarche (OR: 1.87; 95% CI, 1.11-3.14), if the participant was a prior patient at the GHESKIO clinics (OR: 2.17; 95% CI, 1.32-3.58), and participating in the school-based program (OR: 4.17; 95% CI, 2.14-8.12) were significantly associated with vaccination completion.”

Comment 5: Line No 72 GHESKIO -Is the vaccine introduced in only patients with Kaposi sarcoma. Mention other healthy participants. 

Response 5: See response 1.

Comment 6: Line 81 Abbreviation for PIH

Response 6: We thank the reviewer for catching this. Lines 66-67 when the word is first used has been updated to include the acronym; see below:

“In 2009, a three-dose HPV vaccination pilot program was launched in Mirebalais, in central Haiti, supported by an initiative of Zanmi Lasante/Partners in Health (PIH).”

Comment 7: Line 102 Methods and materials, Define Patient /study participants

Response 7: See response 1. 

Comment 8: Line 178 whereas only around one fifth (21.4%) were pregnant at the time of vaccination, all of whom received both doses of the vaccine. HPV vaccine is not administered in pregnant women. Justify this.

Response 8: We thank the reviewer for catching this. That number was an error resulting from our internal review process and was in reference to participants from neighborhoods within 1km of the GHESKIO clinic. See below (line 179-180): 

“More than half (58.8%) of the non-school cohort were from neighborhoods within 1 kilometer of the GHESKIO clinic, whereas only around one fifth (21.4%) from neighborhoods within 1 kilometer of the GHESKIO clinic in the school cohort.“

Comment 9: Line 228 After adjusting for participant age, education, GHESKIO patient status-Authors have to define GHESKIO patient status

Response 9: See response 1. Additionally, we have updated the language to make clear that we are referring to whether the participant was a former patient at a GHESKIO clinic (lines 229-233). See below: 

“After adjusting for participant age, education, whether the participant was a GHESKIO patient or not, cohort (school or non-school), neighborhood distance from GHESKIO clinic, and guardian characteristics, girls who had experienced menarche had 1.87 times higher odds of completing the vaccination series than those who had not experienced menarche (95% confidence interval [CI], 1.11-3.14; P=0.019; Table 2).”

---

## [Decision Letter · Decision Letter 2]

14 May 2021

Success of Community Approach to HPV Vaccination in School-Based and Non-School-Based Settings in Haiti

PONE-D-20-36531R2

Dear Dr. Bane,

We’re pleased to inform you that your manuscript has been judged scientifically suitable for publication and will be formally accepted for publication once it meets all outstanding technical requirements.

Kind regards,

Orvalho Augusto, MD, MPH

Academic Editor

PLOS ONE

Additional Editor Comments (optional):

Reviewers' comments:

Reviewer's Responses to Questions

**Comments to the Author**

1. If the authors have adequately addressed your comments raised in a previous round of review and you feel that this manuscript is now acceptable for publication, you may indicate that here to bypass the “Comments to the Author” section, enter your conflict of interest statement in the “Confidential to Editor” section, and submit your "Accept" recommendation.

Reviewer #2: All comments have been addressed

2. Is the manuscript technically sound, and do the data support the conclusions?

Reviewer #2: Yes

3. Has the statistical analysis been performed appropriately and rigorously? 

Reviewer #2: Yes

4. Have the authors made all data underlying the findings in their manuscript fully available?

Reviewer #2: Yes

5. Is the manuscript presented in an intelligible fashion and written in standard English?

Reviewer #2: Yes

6. Review Comments to the Author

Reviewer #2: Authors have modified the research article and have addressed the concerns. Now the manuscript can be accepted

7. PLOS authors have the option to publish the peer review history of their article (what does this mean?). If published, this will include your full peer review and any attached files.

Reviewer #2: **Yes: **Dr Sabeena Sasidharan Pillai

---

## [Editor Report · Acceptance letter]

16 Jun 2021

PONE-D-20-36531R2 

Success of community approach to HPV vaccination in school-based and non-school-based settings in Haiti 

Dear Dr. Bane:

I'm pleased to inform you that your manuscript has been deemed suitable for publication in PLOS ONE. Congratulations! Your manuscript is now with our production department. 

Kind regards, 

on behalf of

Dr. Orvalho Augusto 

Academic Editor

PLOS ONE